# DEEP ACTIVE LEARNING FOR NAMED ENTITY RECOGNITION

**Yanyao Shen**
UT Austin
Austin, TX 78712
shenyanyao@utexas.edu

**Hyokun Yun**
Amazon Web Services
Seattle, WA 98101
yunhyoku@amazon.com

**Zachary C. Lipton**
Amazon Web Services
Seattle, WA 98101
liptoz@amazon.com

**Yakov Kronrod**
Amazon Web Services
Seattle, WA 98101
kronrod@amazon.com

**Animashree Anandkumar**
Amazon Web Services
Seattle, WA 98101
anima@amazon.com

## ABSTRACT

Deep learning has yielded state-of-the-art performance on many natural language processing tasks including named entity recognition (NER). However, this typically requires large amounts of labeled data. In this work, we demonstrate that the amount of labeled training data can be drastically reduced when deep learning is combined with active learning. While active learning is sample-efficient, it can be computationally expensive since it requires iterative retraining. To speed this up, we introduce a lightweight architecture for NER, *viz.,* the CNN-CNN-LSTM model consisting of convolutional character and word encoders and a long short term memory (LSTM) tag decoder. The model achieves nearly state-of-the-art performance on standard datasets for the task while being computationally much more efficient than best performing models. We carry out incremental active learning, during the training process, and are able to nearly match state-of-the-art performance with just 25% of the original training data.

## 1 INTRODUCTION

Over the past few years, papers applying deep neural networks (DNNs) to the task of named entity recognition (NER) have successively advanced the state-of-the-art (Collobert et al., 2011; Huang et al., 2015; Lample et al., 2016; Chiu & Nichols, 2016; Yang et al., 2016). However, under typical training procedures, the advantages of deep learning diminish when working with small datasets. For instance, on the OntoNotes-5.0 English dataset, whose training set contains 1,088,503 words, a DNN model outperforms the best shallow model by 2.24% as measured by F1 score (Chiu & Nichols, 2016). However, on the comparatively small CoNLL-2003 English dataset, whose training set contains 203,621 words, the best DNN model enjoys only a 0.4% advantage. To make deep learning more broadly useful, it is crucial to reduce its training data requirements.

Generally, the annotation budget for labeling is far less than the total number of available (unlabeled) samples. For NER, getting unlabeled data is practically free, owing to the large amount of content that can be efficiently scraped off the web. On the other hand, it is especially expensive to obtain annotated data for NER since it requires multi-stage pipelines with sufficiently well-trained annotators (Kilicoglu et al., 2016; Bontcheva et al., 2017). In such cases, active learning offers a promising approach to efficiently select the set of samples for labeling. Unlike the supervised learning setting, in which examples are drawn and labeled at random, in the active learning setting, the algorithm can choose which examples to label.

Active learning aims to select a *more informative* set of examples in contrast to supervised learning, which is trained on a set of randomly drawn examples. A central challenge in active learning is to determine what constitutes *more informative* and how the active learner can recognize this based on what it already knows. The most common approach is *uncertainty sampling*, in which the model preferentially selects examples for which it's current prediction is least confident. Other approaches

include representativeness-based sampling where the model selects a diverse set that represent the input space without adding too much redundancy.

**In this work**, we investigate practical active learning algorithms on lightweight deep neural network architectures for the NER task. Training with active learning proceeds in multiple rounds. Traditional active learning schemes are expensive for deep learning since they require complete retraining of the classifier with newly annotated samples after each round. In our experiments, for example, the model must be retrained 54 times. Because retraining from scratch is not practical, we instead carry out incremental training with each batch of new labels: we mix newly annotated samples with the older ones, and update our neural network weights for a small number of epochs, before querying for labels in a new round. This modification drastically reduces the computational requirements of active learning methods and makes it practical to deploy them.

We further reduce the computational complexity by selecting a lightweight architecture for NER. We propose a new CNN-CNN-LSTM architecture for NER consisting of a convolutional character-level encoder, convolutional word-level encoder, and long short term memory (LSTM) tag decoder. This model handles out-of-vocabulary words gracefully and, owing to the greater reliance on convolutions (vs recurrent layers), trains much faster than other deep models while performing competitively.

We introduce a simple uncertainty-based heuristic for active learning with sequence tagging. Our model selects those sentences for which the length-normalized log probability of the current prediction is the lowest. Our experiments with the Onto-Notes 5.0 English and Chinese datasets demonstrate results comparable to the Bayesian active learning by disagreement method (Gal et al., 2017). Moreover our heuristic is faster to compute since it does not require multiple forward passes. On the OntoNotes-5.0 English dataset, our approach matches 99% of the F1 score achieved by the best deep models trained in a standard, supervised fashion despite using only a 24.9% of the data. On the OntoNotes-5.0 Chinese dataset, we match 99% performance with only 30.1% of the data. Thus, we are able to achieve state of art performance with drastically lower number of samples.

## 2 RELATED WORK

**Deep learning for named entity recognition**    The use of DNNs for NER was pioneered by Collobert et al. (2011), who proposed an architecture based on temporal convolutional neural networks (CNNs) over the sequence of words. Since then, many papers have proposed improvements to this architecture. Huang et al. (2015) proposed to replace CNN encoder in Collobert et al. (2011) with bidirectional LSTM encoder, while Lample et al. (2016) and Chiu & Nichols (2016) introduced hierarchy in the architecture by replacing hand-engineered character-level features in prior works with additional bidirectional LSTM and CNN encoders respectively. In other related work, Mesnil et al. (2013) and Nguyen et al. (2016) pioneered the use of recurrent neural networks (RNNs) for decoding tags. However, most recent competitive approaches rely upon CRFs as decoder (Lample et al., 2016; Chiu & Nichols, 2016; Yang et al., 2016). In this work, we demonstrate that LSTM decoders outperform CRF decoders and are faster to train when the number of entity types is large.

**Active learning**    While learning-theoretic properties of active learning algorithms are well-studied (Dasgupta et al., 2005; Balcan et al., 2009; Awasthi et al., 2014; Yan & Zhang, 2017; Beygelzimer et al., 2009), classic algorithms and guarantees cannot be generalized to DNNs, which are currently are the state-of-the-art techniques for NER. Owing to the limitations of current theoretical analysis, more practical active learning applications employ a range of heuristic procedures for selecting examples to query. For example, Tong & Koller (2001) suggests a margin-based selection criteria, while Settles & Craven (2008) while Shen et al. (2004) combines multiple criteria for NLP tasks. Culotta & McCallum (2005) explores the application of least confidence criterion for linear CRF models on sequence prediction tasks. For a more comprehensive review of the literature, we refer to Settles (2010) and Olsson (2009).

**Deep active learning**    While DNNs have achieved impressive empirical results across diverse applications (Krizhevsky et al., 2012; Hinton et al., 2012; Manning, 2016), active learning approaches for these models have yet to be well studied, and most current work addresses image classification. Wang et al. (2016) claims to be the first to study active learning for image classification with CNNs and proposes methods based on uncertainty-based sampling, while Gal et al. (2017) and Kendall & Gal (2017) show that sampling based on a Bayesian uncertainty measure can be more advantageous. In one related paper, Zhang et al. (2017) investigate active learning for sentence classification with

| Character-Level Encoder | Word-Level Encoder | Tag Decoder | Reference |
|---|---|---|---|
| None | CNN | CRF | Collobert et al. (2011) |
| None | RNN | RNN | Mesnil et al. (2013) |
| None | RNN | GRU | Nguyen et al. (2016) |
| None | LSTM | CRF | Huang et al. (2015) |
| LSTM | LSTM | CRF | Lample et al. (2016) |
| CNN | LSTM | CRF | Chiu & Nichols (2016) |
| CNN | LSTM | LSTM, Pointer Networks | Zhai et al. (2017) |
| GRU | GRU | CRF | Yang et al. (2016) |
| None | Dilated CNN | Independent Softmax, CRF | Strubell et al. (2017) |
| CNN | CNN | LSTM | Ours |

Table 1: Prior works on neural architectures for sequence tagging, and their corresponding design choices.

CNNs. However, to our knowledge, prior to this work, deep active learning for sequence tagging tasks, which often have structured output space and variable-length input, has not been studied.

## 3    NER MODEL DESCRIPTION

Most active learning methods require frequent retraining of the model as new labeled examples are acquired. Therefore, it is crucial that the model can be efficiently retrained. On the other hand, we would still like to reach the level of performance rivaling state-of-the-art DNNs.

To accomplish this, we first identify that many DNN architectures for NER can be decomposed into three components: 1) the character-level encoder, which extracts features for each word from characters, 2) the word-level encoder which extracts features from the surrounding sequence of words, and 3) the tag decoder, which induces a probability distribution over any sequences of tags. This conceptual framework allows us to view a variety of DNNs in a unified perspective; see Table 1.

Owing to the superior computational efficiency of CNNs over LSTMs, we propose a lightweight neural network architecture for NER, which we name CNN-CNN-LSTM and describe below.

| Formatted Sentence | [BOS] | Kate | lives | on | Mars | [EOS] | [PAD] |
|---|---|---|---|---|---|---|---|
| Tag | O | S-PER | O | O | S-LOC | O | O |

Table 2: Example formatted sentence. To avoid clutter, [BOW] and [EOW] symbols are not shown.

**Data Representation**     We represent each input sentence as follows; First, special [BOS] and [EOS] tokens are added at the beginning and the end of the sentence, respectively. In order to batch the computation of multiple sentences, sentences with similar length are grouped together into buckets, and [PAD] tokens are added at the end of sentences to make their lengths uniform inside of the bucket. We follow an analogous procedure to represent the characters in each word. For example, the sentence 'Kate lives on Mars' is formatted as shown in Table 2. The formatted sentence is denoted as $\{\mathbf{x}_{ij}\}$, where $\mathbf{x}_{ij}$ is the one-hot encoding of the $j$-th character in the $i$-th word.

**Character-Level Encoder**     For each word $i$, we use CNNs (LeCun et al., 1995) to extract character-level features $\mathbf{w}_i^{\text{char}}$ (Figure 1). While LSTM recurrent neural network (Hochreiter & Schmidhuber, 1997) slightly outperforms CNN as a character-level encoder, the improvement is not statistically significant and the computational cost of LSTM encoders is much higher than CNNs (see Section 5, also Reimers & Gurevych (2017) for detailed analysis).

We apply ReLU nonlinearities (Nair & Hinton, 2010) and dropout (Srivastava et al., 2014) between CNN layers, and include a residual connection between input and output of each layer (He et al., 2016). So that our representation of the word is of fixed length, we apply max-pooling on the outputs of the topmost layer of the character-level encoder (Kim, 2014).

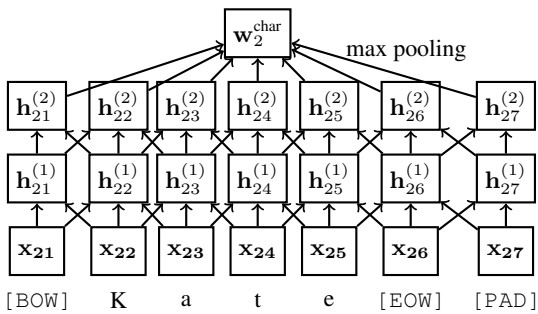

Figure 1: Example CNN architecture for Character-level Encoder with two layers.

**Word-Level Encoder**     To complete our representation of each word, we concatenate its character-level features with $\mathbf{w}_i^{\text{emb}}$, a latent word embedding corresponding to that word:

$$\mathbf{w}_i^{\text{full}} := \left( \mathbf{w}_i^{\text{char}}, \mathbf{w}_i^{\text{emb}} \right).$$

We initialize the latent word embeddings with with word2vec training Mikolov et al. (2013) and then update the embeddings over the course of training. In order to generalize to words unseen in the training data, we replace each word with a special [UNK] (unknown) token with 50% probability during training, an approach that resembles the word-drop method due to Lample et al. (2016).

Given the sequence of word-level input features $\mathbf{w}_1^{\text{full}}, \mathbf{w}_2^{\text{full}}, \ldots, \mathbf{w}_n^{\text{full}}$, we extract word-level representations $\mathbf{h}_1^{\text{Enc}}, \mathbf{h}_2^{\text{Enc}}, \ldots, \mathbf{h}_n^{\text{Enc}}$ for each word position in the sentence using a CNN. In Figure 2, we depict an instance of our architecture with two convolutional layers and kernels of width 3. We concatenate the representation at the $l$-th convolutional layer $\mathbf{h}_i^{(l)}$, with the input features $\mathbf{w}_i^{\text{full}}$:

$$\mathbf{h}_i^{\text{Enc}} = \left( \mathbf{h}_i^{(l)}, \mathbf{w}_i^{\text{full}} \right)$$

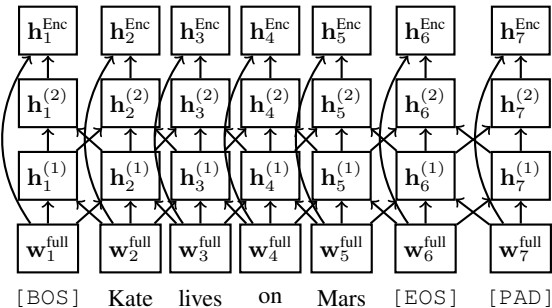

Figure 2: Example CNN architecture for Word-level Encoder with two layers.

LSTM RNNs can also perform word-level encoding Huang et al. (2015), and models with LSTM word-level encoding give a slight (but not significant) boost over CNN word-level encoders in terms of F1 score (see Section 5). However, CNN word-level encoders are considerably faster (Strubell et al., 2017), which is crucial for the iterative retraining in our active learning scheme.

**Tag Decoder**     The tag decoder induces a probability distribution over sequences of tags, conditioned on the word-level encoder features: $\mathbb{P}\left[ y_2, y_3, \ldots, y_{n-1} \mid \left\{ \mathbf{h}_i^{\text{Enc}} \right\} \right]$[1]. Chain CRF (Lafferty et al., 2001) is a popular choice for tag decoder, adopted by most modern DNNs for NER:

$$\mathbb{P}\left[ t_2, t_3, \ldots, t_{n-1} \mid \left\{ \mathbf{h}_i^{\text{Enc}} \right\} \right] \propto \exp\left( \sum_{i=2}^{n-1} \left\{ W\mathbf{h}_i^{\text{Enc}} + b \right\}_{t_i} + A_{t_{i-1}, t_i} \right), \qquad (1)$$

---

[1]$y_1$ and $y_n$ are ignored because they correspond to auxiliary words [BOS] and [EOS]. If [PAD] words are introduced, they are ignored as well.

where $W$, $A$, $b$ are learnable parameters, and $\{\cdot\}_{t_i}$ refers to the $t_i$-th coordinate of the vector. To compute the partition function of (1), which is required for training, usually dynamic programming is employed, and its time complexity is $O(nT^2)$ where $T$ is the number of entity types (Collobert et al., 2011).

Alternatively, we use an LSTM RNN for the tag decoder, as depicted in Figure 3. At the first time step, the [GO]-symbol is provided as $y_1$ to the decoder LSTM. At each time step $i$, the LSTM decoder computes $\mathbf{h}_{i+1}^{\text{Dec}}$, the hidden state for decoding word $i + 1$, using the last tag $y_i$, the current decoder hidden state $\mathbf{h}_i^{\text{Dec}}$, and the learned representation of next word $\mathbf{h}_{i+1}^{\text{Enc}}$. Using a softmax loss function, $y_{i+1}$ is decoded; this is further fed as an input to the next time step.

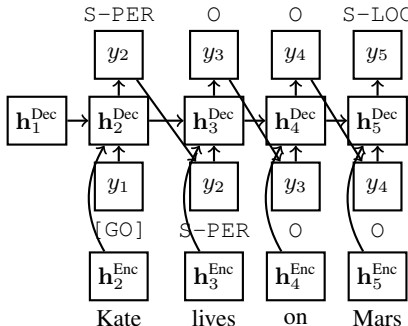

Figure 3: LSTM architecture for Tag Decoder.

Since this is a locally normalized model (Andor et al., 2016), it does not require the costly computation of partition function, and it allows us to significantly speed up training compared to using CRFs. Also, we observed that while it is computationally intractable to find the best sequence of tags with an LSTM decoder, greedily decoding tags from left to right yields the performance comparable to chain CRF decoder (see Appendix A). While the use of RNNs tag decoders has been explored (Mesnil et al., 2013; Nguyen et al., 2016; Zhai et al., 2017), we demonstrate for the first time that models using RNNs instead of CRFs for tag decoder can achieve state-of-the-art performance. See Section 5.

## 4 ACTIVE LEARNING

Labeling data for NER usually requires manual annotations by human experts, which are costly to acquire at scale. Active learning seeks to ameliorate this problem by strategically choosing which examples to annotate, in the hope of getting greater performance with fewer annotations. To this end, we consider the following setup for interactively acquiring annotations. The learning process consists of multiple rounds: At the beginning of each round, the active learning algorithm chooses sentences to be annotated up to the predefined budget. After receiving annotations, we update the model parameters by training on the augmented dataset, and proceeds to the next round. We assume that the cost of annotating a sentence is proportional to the number of words in the sentence and that every word in the selected sentence must be annotated at once, i.e. we do not allow or account for partially annotated sentences.

While various existing active learning strategies suit this setup (Settles, 2010), we explore the uncertainty sampling strategy. With the uncertainty-based sampling strategy (Lewis & Gale, 1994), we rank the unlabeled examples according to the current model's uncertainty in its prediction of the corresponding labels. We consider three ranking methods, each of which can be easily implemented in the CNN-CNN-LSTM model or most other deep neural approaches to NER.

**Least Confidence (LC):** Culotta & McCallum (2005) proposed to sort examples in ascending order according to the probability assigned by the model to the most likely sequence of tags:

$$1 - \max_{y_1, \ldots, y_n} \mathbb{P}\left[y_1, \ldots, y_n \mid \{\mathbf{x}_{ij}\}\right]. \tag{2}$$

Exactly computing (2) requires identifying the most likely sequence of tags according to the LSTM decoder. Because determining the most likely sequence is intractable, we approximate the score by using the probability assigned to the greedily decoded sequence.

**Maximum Normalized Log-Probability (MNLP):** Preliminary analysis revealed that the LC method disproportionately selects longer sentences. Note that sorting unlabeled examples in descending order by (2) is equivalent to sorting in ascending order by the following scores:

$$\max_{y_1,\ldots,y_n} \mathbb{P}\left[y_1,\ldots,y_n \mid \{\mathbf{x}_{ij}\}\right] \Leftrightarrow \max_{y_1,\ldots,y_n} \prod_{i=1}^{n} \mathbb{P}\left[y_i \mid y_1,\ldots,y_{n-1}, \{\mathbf{x}_{ij}\}\right]$$

$$\Leftrightarrow \max_{y_1,\ldots,y_n} \sum_{i=1}^{n} \log \mathbb{P}\left[y_i \mid y_1,\ldots,y_{n-1}, \{\mathbf{x}_{ij}\}\right]. \tag{3}$$

Since (3) contains summation over words, LC method naturally favors longer sentences. Because longer sentences requires more labor for annotation, we find this undesirable, and propose to normalize (3) as follows, which we call Maximum Normalized Log-Probability method:

$$\max_{y_1,\ldots,y_n} \frac{1}{n} \sum_{i=1}^{n} \log \mathbb{P}\left[y_i \mid y_1,\ldots,y_{n-1}, \{\mathbf{x}_{ij}\}\right].$$

**Bayesian Active Learning by Disagreement (BALD):** We also consider sampling according to the measure of uncertainty proposed by Gal et al. (2017). Observing a correspondence between dropout (Srivastava et al., 2014) and deep Gaussian processes (Damianou & Lawrence, 2013), they propose that the variability of the predictions over successive forward passes due to dropout can be interpreted as a measure of the model's uncertainty (Gal & Ghahramani, 2016). Denote $\mathbb{P}^1, \mathbb{P}^2, \ldots \mathbb{P}^M$ as models resulting from applying $M$ independently sampled dropout masks. One measure of our uncertainty on the $i$th word is $f_i$, the fraction of models which disagreed with the most popular choice:

$$f_i = 1 - \frac{\max_y \left|\left\{m : \mathrm{argmax}_{y'} \mathbb{P}^m\left[y_i = y'\right] = y\right\}\right|}{M},$$

where $|\cdot|$ denotes cardinality of a set. We normalize this by the number of words as $\frac{1}{n}\sum_{j=1}^{n} f_j$, In this paper, we draw $M = 100$ independent dropout masks.

**Other Sampling Strategies.** Consider that the confidence of the model can help to distinguish between hard and easy samples. Thus, sampling examples where the model is uncertain might save us from sampling too heavily from regions where the model is already proficient. But intuitively, when we query a batch of examples in each round, we might want to guard against querying examples that are too similar to each other, thus collecting redundant information. We also might worry that a purely uncertainty-based approach would oversample outliers. Thus we explore techniques to guard against these problems by selecting a set of samples that is *representative* of the dataset. Following Wei et al. (2015), we express the problem of maximizing representativeness of a labeled set as a submodular optimization problem, and provide an efficient streaming algorithm adapted to use a constraint suitable to the NER task.

Our approach to representativeness-based sampling proceeds as follows: Denote $\mathbb{X}$ as the set of all samples, and $\mathbb{X}^L, \mathbb{X}^U$ representing the set of labeled and unlabeled samples respectively. For an unlabeled set $\mathbb{S} \in \mathbb{X}^U$, the utility $f_w$ is defined as the summation of marginal utility gain over all unlabeled points, weighted by their uncertainty. More formally,

$$f_w(\mathbb{S}) = \sum_{i \in \mathbb{X}^U} \mathrm{US}(i) \cdot \left[\max_{j \in \mathbb{S} \cup \mathbb{X}^L} w(i,j) - \max_{j \in \mathbb{X}^L} w(i,j)\right], \tag{4}$$

where $\mathrm{US}(i)$ is the uncertainty score on example $i$. In order to find a good set $\mathbb{S}$ with high $f_w$ value, we exploit the submodularity of the function, and use an online algorithm under knapsack constraint. More details of this method can be found in the supplementary material (Appendix C). In our experiments, this approach fails to match the uncertainty-based heuristics or to improve upon them when used in combination. Nevertheless, we describe the algorithm and include the negative results for their scientific value.

## 5 EXPERIMENTS

### 5.1 MODEL EFFICIENCY AND PERFORMANCE

In order to evaluate the efficiency and performance CNN-CNN-LSTM as well as other alternatives, we run the experiments on two widely used NER datasets: CoNLL-2003 English (Tjong Kim Sang & De Meulder, 2003) and OntoNotes-5.0 English (Pradhan et al., 2013). We use the standard split of training/validation/test sets, and use the validation set performance to determine hyperparameters such as the learning rate or the number of iterations for early stopping. Unlike Lample et al. (2016) and Chiu & Nichols (2016), we do *not* train on the validation dataset. We report the F1 score for each model, which is standard. We only consider neural models in this comparison, since they outperform non-neural models for this task. Since our goal here is to compare neural network architectures, we did not experiment with gazetteers.

For neural architectures previously explored by others, we simply cite reported metrics. For LSTM word-level encoder, we use single-layer model with 100 hidden units for CoNLL-2003 English (following Lample et al. (2016)) and two-layer model with 300 hidden units for OntoNotes 5.0 datasets (following Chiu & Nichols (2016)). For character-level LSTM encoder, we use single-layer LSTM with 25 hidden units (following Lample et al. (2016)). For CNN word-level encoder, we use two-layer CNNs with 800 filters and kernel width 5, and for CNN character-level encoder, we use single-layer CNNs with 50 filters and kernel width 3 (following Chiu & Nichols (2016)). Dropout probabilities are all set as 0.5. We use structured skip-gram model (Ling et al., 2015) trained on Gigawords-English corpus (Graff & Cieri, 2003), which showed a good boost over vanilla skip-gram model (Mikolov et al., 2013) we do not report here. We use vanilla stochastic gradient descent, since it is commonly reported in the named entity recognition literature that this outperforms more sophisticated methods at convergence (Lample et al., 2016; Chiu & Nichols, 2016). We uniformly set the step size as 0.001 and the batch size as 128. When using LSTMs for the tag decoder, for inference, we only use greedy decoding; beam search gave very marginal improvement in our initial experiments. We repeat each experiment four times, and report mean and standard deviation. In terms of measuring the training speed of our models, we compute the time spent for one iteration of training on the dataset, with eight K80 GPUs in `p2.8xlarge` on Amazon Web Services[2].

Table 3 and Table 4 show the comparison between our model and other best performing models. LSTM tag decoder shows performance comparable to CRF tag decoder, and it works better than the CRF decoder when used with CNN encoder; compare CNN-CNN-LSTM vs. CNN-CNN-CRF on both tables. On the CoNLL-2003 English dataset which has only four entity types, the training speed of CNN-CNN-LSTM and CNN-CNN-CRF are comparable. However, on the OntoNotes 5.0 English dataset which has 18 entity types, the training speed of CNN-CNN-LSTM is twice faster than CNN-CNN-CRF because the time complexity of computing the partition function for CRF is quadratic to the number of entity types. CNN-CNN-LSTM is also 44% faster than CNN-LSTM-LSTM on OntoNotes, showing the advantage of CNN over LSTM as word encoder; on CoNLL-2003, sentences tend to be shorter and this advantage was not clearly seen; its median number of words in sentences is 12 opposed 17 of OntoNotes. Compared to the CNN-LSTM-CRF model, which is considered as a state-of-the-art model in terms of performance (Chiu & Nichols, 2016; Strubell et al., 2017), CNN-CNN-LSTM provides four times speedup in terms of the training speed, and achieves comparatively high performance measured by F1 score.

### 5.2 PERFORMANCE OF DEEP ACTIVE LEARNING

We use OntoNotes-5.0 English and Chinese data (Pradhan et al., 2013) for our experiments. The training datasets contain 1,088,503 words and 756,063 words respectively. State-of-the-art models trained on the full training sets achieve F1 scores of 86.86 Strubell et al. (2017) and 75.63 (our CNN-CNN-LSTM) on the test sets.

**Comparisons of selection algorithms**    We empirically compare selection algorithms proposed in Section 4, as well as uniformly random baseline (**RAND**). All algorithms start with an identical 1% of original training data and a randomly initialized model. In each round, every algorithm chooses sentences from the rest of the training data until 20,000 words have been selected, adding this data to

---

[2]https://aws.amazon.com/ec2/instance-types/p2/

| Char | Word | Tag | Reference | F1 | Sec/Epoch |
|------|------|-----|-----------|-----|-----------|
| None | CNN | CRF | Collobert et al. (2011) | 88.67 | - |
| None | LSTM | CRF | Huang et al. (2015) | 90.10 | - |
| LSTM | LSTM | CRF | Lample et al. (2016) | 90.94 | - |
| CNN | LSTM | CRF | Chiu & Nichols (2016) | 90.91 ± 0.20 | - |
| GRU | GRU | CRF | Yang et al. (2016) | 90.94 | - |
| None | Dilated CNN | CRF | Strubell et al. (2017) | 90.54 ± 0.18 | - |
| LSTM | LSTM | LSTM | | 90.89 ± 0.19 | 49 |
| CNN | LSTM | LSTM | | 90.58 ± 0.28 | 11 |
| CNN | CNN | LSTM | | 90.69 ± 0.19 | 11 |
| CNN | CNN | CRF | | 90.35 ± 0.24 | 12 |

Table 3: Evaluations on the test set of CoNLL-2003 English

| Char | Word | Tag | Reference | F1 | Sec/Epoch |
|------|------|-----|-----------|-----|-----------|
| CNN | LSTM | CRF | Chiu & Nichols (2016) | 86.28 ± 0.26 | 83* |
| None | Dilated CNN | CRF | Strubell et al. (2017) | 86.84 ± 0.19 | - |
| CNN | LSTM | LSTM | | 86.40 ± 0.48 | 76 |
| CNN | CNN | LSTM | | 86.52 ± 0.25 | 22 |
| CNN | CNN | CRF | | 86.15 ± 0.08 | 44 |
| LSTM | LSTM | LSTM | | 86.63 ± 0.49 | 206 |

Table 4: Evaluations on the test set of OntoNotes 5.0 English. (*) Training speed of CNN-LSTM-CRF model was measured with our own implementation of it.

its training set. We then update each model by stochastic gradient descent on its augmented training dataset for 50 passes. We evaluate the performance of algorithm by its F1 score on the test dataset.

Figure 4 shows the results. All active learning algorithms perform significantly better than the random baseline. Among active learners, **MNLP** and **BALD** slightly outperformed traditional **LC** in early rounds. Note that **MNLP** is computationally more efficient than **BALD**, since it only requires a single forward pass on the unlabeled dataset to compute uncertainty scores, whereas **BALD** requires multiple forward passes. Impressively, active learning algorithms achieve 99% performance of the best deep model trained on full data using only 24.9% of the training data on the English dataset and 30.1% on Chinese. Also, 12.0% and 16.9% of training data were enough for deep active learning algorithms to surpass the performance of the shallow models from Pradhan et al. (2013) trained on the full training data. We repeated the experiment eight times and confirmed that the trend is replicated across multiple runs; see Appendix B for details.

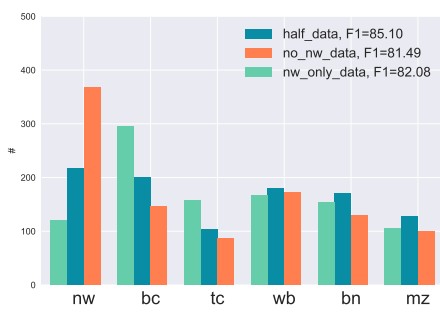

Figure 5: Genre distribution of top 1,000 sentences chosen by an active learning algorithm

**Detection of under-explored genres** To better understand how active learning algorithms choose informative examples, we designed the following experiment. The OntoNotes datasets consist of six genres: broadcast conversation (bc), braodcast news (bn), magazine genre (mz), newswire (nw), telephone conversation (tc), weblogs (wb). We created three training datasets: *half-data*, which contains random 50% of the original training data, *nw-data*, which contains sentences only from newswire (51.5% of words in the original data), and *no-nw-data*, which is the complement of *nw-data*. Then, we trained CNN-CNN-LSTM model on each dataset. The model trained on *half-data*

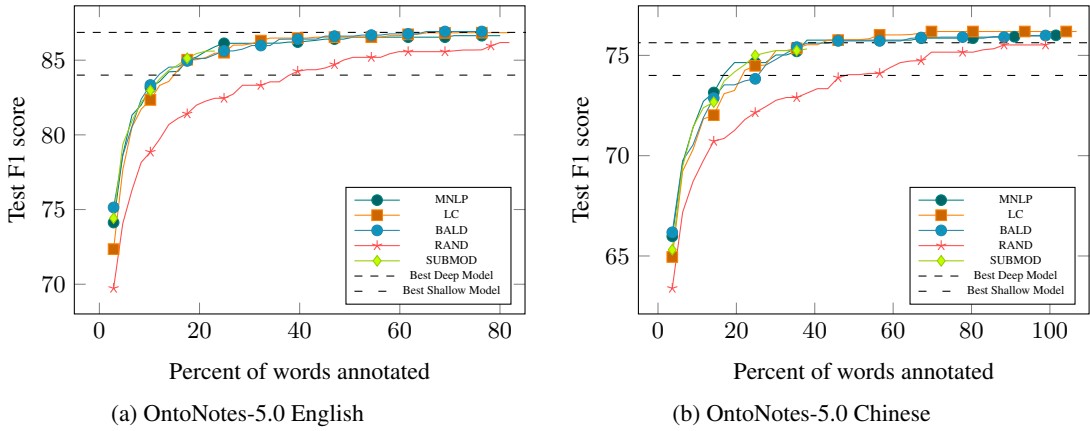

(a) OntoNotes-5.0 English       (b) OntoNotes-5.0 Chinese

Figure 4: F1 score on the test dataset, in terms of the number of words labeled.

achieved 85.10 F1, significantly outperforming others trained on biased datasets (*no-nw-data*: 81.49, *nw-only-data*: 82.08). This showed the importance of good genre coverage in training data. Then, we analyzed the genre distribution of 1,000 sentences **MNLP** chose for each model (see Figure 5). For *no-nw-data*, the algorithm chose many more newswire (nw) sentences than it did for unbiased *half-data* (367 vs. 217). On the other hand, it undersampled newswire sentences for *nw-only-data* and increased the proportion of broadcast news and telephone conversation, which are genres distant from newswire. Impressively, although we did not provide the genre of sentences to the algorithm, it was able to automatically detect underexplored genres.

## 6 Conclusion

We proposed an efficient model for NER tasks which gives high performances on well-established datasets. Further, we use deep active learning algorithms for NER and empirically demonstrated that they achieve state-of-the-art performance with much less data than models trained in the standard supervised fashion.

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

## A    EFFECT OF BEAM SIZE ON LSTM DECODER

One potential concern when decoding with an LSTM decoder as compared to using a CRF decoder is that finding the best sequence of labels that maximizes the probability $\mathbb{P}\left[t_2, t_3, \ldots, t_{n-1} \mid \left\{\mathbf{h}_i^{\text{Enc}}\right\}\right]$ is computationally intractable. In practice, however, we find that simple greedy decoding, i.e., beam search with beam size 1, works surprisingly well. Table 5 shows how changing the beam size of decoder affects the performance of the model. It can be seen that the performance of the model changes very little with respect to the beam size. Beam search with size 2 is marginally better than greedy decoding, and further increasing the beam size did not help at all. Moreover, we note that while it may be computationally efficient to pick the most likely tag sequence given a CRF encoder, the LSTM decoder may give more accurate predictions, owing to it's greater representational power and ability to model long-range dependencies. Thus even if we do not always choose the most probable tag sequence from the LSTM, we can still outperform the CRF (as our experiments demonstrate).

| Beam Size | F1 |
|:---:|:---:|
| 1 | 87.26 |
| 2 | 87.34 |
| 4 | 87.33 |
| 8 | 87.33 |
| 16 | 87.33 |

Table 5: Effect of beam size in LSTM decoder. We used a single LSTM-LSTM-LSTM model, and evaluated on OntoNotes 5.0 English dataset.

## B    LEARNING CURVE IN ACTIVE LEARNING EXPERIMENTS ACROSS MULTIPLE RUNS

In order to understand the variability of learning curves in Figure 4a across experiments, we repeated the active learning experiment on OntoNotes-5.0 English eight times, each of which started with different initial dataset chosen randomly. Figure 6 shows the result in first nine rounds of labeled data acquisition. While MNLP, LC and BALD are all competitive against each other, there is a noticeable trend that MNLP and BALD outperforms LC in early rounds of data acquisition.

## C    REPRESENTATIVENESS-BASED ACTIVE LEARNING

Consider that the confidence of the model can help to distinguish between hard and easy samples. Thus, sampling examples where the model is uncertain might save us from sampling too heavily from regions where the model is already proficient. But intuitively, when we query a batch of examples in each round, we might want to guard against querying examples that are too similar to each other, thus collecting redundant information. We also might worry that a purely uncertainty-based approach would oversample outliers. Thus we explore techniques to guard against these problems by selecting a set of samples that is *representative* of the dataset. Following Wei et al. (2015), we express the problem of maximizing representativeness of a labeled set as a submodular optimization problem, and provide an efficient streaming algorithm adapted to use a constraint suitable to the NER task. We also provide some with theoretical guarantees.

**Submodular utility function**    In order to reason about the similarity between samples, we first embed each sample $i$ into a fixed-dimensional euclidean space as a vector $\mathbf{x}_i$. We consider two embedding methods: 1) the average of pre-trained word embeddings, $\frac{1}{n}\sum_{t=1}^{n} \mathbf{w}_t^{\text{emb}}$, which can be efficiently computed without training of the NER model, and 2) the average of activation maps at the topmost layer of the encoder $\frac{1}{n}\sum_{t=1}^{n} \mathbf{h}_t^{\text{Enc}}$, an embedding which might be better suited to the context of the NER task. Then, we consider the following options for defining similarity scores $w(i, j)$ between each pair of samples $i$ and $j$: $w(i, j) = d - \|x^i - x^j\|_p$ where $d = \max_{i,j \in \mathbb{X}} \|x^i - x^j\|_p$

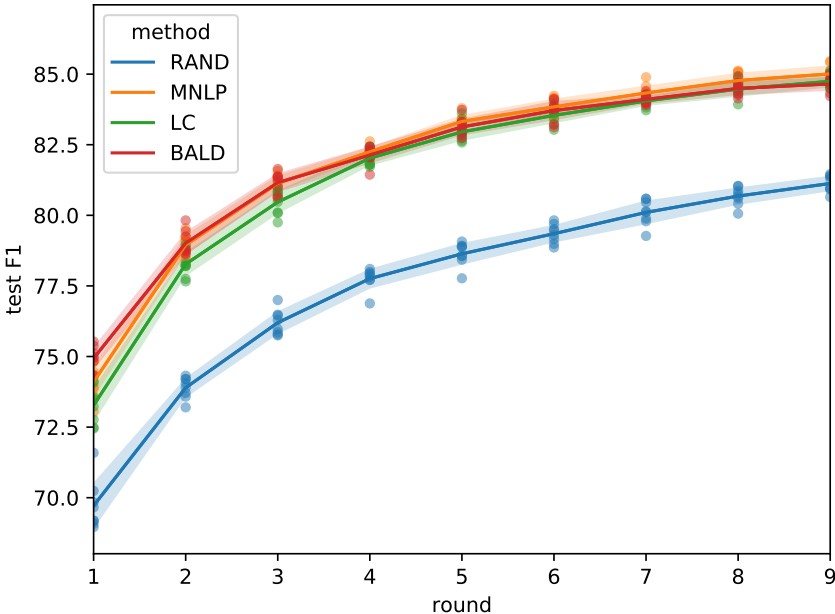

Figure 6: Test F1 score in first nine rounds of labeled data acquisition on OntoNotes-5.0 English dataset across multiple runs of the active learning experiment. The experiment was repeated eight times, with different initial dataset. Error bars indicates standard deviation, and dots indicate individual observation.

for $p = 1, 2$ which corresponds to closeness in $L_1$ and $L_2$ distance (Wei et al., 2015), and $w(i, j) = 1 + \frac{x_i \cdot x_j}{\|x_i\| \cdot \|x_j\|}$, which corresponds to cosine similarity.

Now, we formally define the utility function for labeling new samples. Denote $\mathbb{X}$ as the set of all samples which can be partitioned into two disjoint sets $\mathbb{X}^{\mathbb{L}}$, $\mathbb{X}^{\mathbb{U}}$ representing labeled and unlabeled samples, respectively. Let $\mathbb{S} \subseteq \mathbb{X}^{\mathbb{U}}$ be a subset of unlabeled samples, then, the utility of labeling the set is defined as follows:

$$f(\mathbb{S}) = \sum_{i \in \mathbb{X}^{\mathbb{U}}} \left[ \max_{j \in \mathbb{S} \cup \mathbb{X}^{\mathbb{L}}} w(i, j) - \max_{j \in \mathbb{X}^{\mathbb{L}}} w(i, j) \right], \tag{5}$$

where the function measures incremental gain of similarity between the labeled set and the rest. Given such utility function $f(\cdot)$, choosing a set $\mathbb{S}$ that maximizes the function within the budget can be seen as a monotone submodular maximization problem under a knapsack constraint (Krause & Golovin, 2012):

$$\max_{\mathbb{S} \subseteq \mathbb{X}^{\mathbb{U}}, \sum_{e \in \mathbb{S}} k(\{e\}) \leq K} f(\mathbb{S}) \tag{6}$$

where $k(\mathbb{S})$ is the budget for the sample set $\mathbb{S}$, and $K$ is the total budget within each round. Note that we need to consider the knapsack constraint instead of the cardinality constraint used in the prior work (Wei et al., 2015), because the entire sentence needs to be labeled once selected and sequences of length confer different labeling costs.

**Combination with uncertainty sampling**  Representation-based sampling can benefit from uncertainty-based sampling in the following two ways. First, we can re-weight each sample in the utility function (5) to reflect current model's uncertainty on it:

$$f_w(\mathbb{S}) = \sum_i \text{US}(i) \cdot \left[ \max_{j \in \mathbb{S} \cup \mathbb{X}^{\mathbb{L}}} w(i, j) - \max_{j \in \mathbb{X}^{\mathbb{L}}} w(i, j) \right], \tag{7}$$

---

**Algorithm 1** Representativeness-based Sampling

1: **Input:** Samples $\{\mathbb{X}^{\mathbb{U}}, \mathbb{X}^{\mathbb{L}}\}$, budget $K$,
   pretrained model $\mathbb{M}$ using $\mathbb{X}^{\mathbb{L}}$
2: **while** Test score of $\mathbb{M}$ less than $th$ **do**
3:     Rank $\mathbb{X}^{\mathbb{U}}$ according to Sec. 4,
       $\tilde{\mathbb{X}}^{\mathbb{U}}$ = top samples $\mathbb{S}$ within budget $t \cdot K$.
4:     Set $f$ according to (5) or (7)
5:     $\mathbb{S} = \mathsf{StreamSubmodMax}(f, \tilde{\mathbb{X}}^{\mathbb{U}})$.
6:     $\{\tilde{\mathbb{X}}^{\mathbb{U}}, \mathbb{X}^{\mathbb{L}}\} = \{\mathbb{X}^{\mathbb{U}} - \mathbb{S}, \mathbb{X}^{\mathbb{L}} \cup \mathbb{S}\}$
7:     Train $\mathbb{M}$ with $\mathbb{X}^{\mathbb{L}}$.
8: **Output:** $\mathbb{M}$

---

**Algorithm 2** StreamSubmodMax

1: **Input:** Submodular function $g$, set $\tilde{\mathbb{X}}^{\mathbb{U}}$
2: $m = \max_{e \in \tilde{\mathbb{X}}^{\mathbb{U}}} g(\{e\})/k(\{e\})$
3: $O = \{(1+\epsilon)^i | i \in \mathbb{Z}, (1+\epsilon)^i \in [m, Km]\}$
4: $\mathbb{S}_v := \emptyset, \forall v \in O$
5: **for** $e$ in $\tilde{\mathbb{X}}^{\mathbb{U}}$ **do**
6:     **for** $v \in O$ and $k(\mathbb{S}_v \cup \{e\}) \leq K$ **do**
7:         **if** $\Delta_g(e|\mathbb{S}_v) \geq \frac{k(\{e\})(v/2 - g(\mathbb{S}_v))}{K - k(\mathbb{S}_v)}$ **then**
8:             $\mathbb{S}_v := \mathbb{S}_v \cup \{e\}$
9: **Output:** $\arg \max_{v \in O} f(\mathbb{S}_v)$

---

where $\mathrm{US}(i)$ is the uncertainty score on example $i$. Second, even with the state-of-the-art submodular optimization algorithms, the optimization problem (6) can be computationally intractable. To improve the computational efficiency, we restrict the set of unlabeled examples to top samples from uncertainty sampling within budget $t \cdot K$, where $t$ is a multiplication factor we set as $4$ in our experiments.

**Streaming algorithm for sample selection**    Even with the reduction of candidates with uncertainty sampling, (6) is still a computationally challenging problem and requires careful design of optimization algorithms. Suppose $l$ is the number of samples we need to consider. In the simplistic case in which all the samples have the same length and thus the knapsack constraint degenerates to the cardinality constraint, the greedy algorithm (Nemhauser et al., 1978) has an $(1 - 1/e)$-approximation guarantee. However, it requires calculating the utility function $O(l^2 n)$ times, where $n$ is the number of unlabeled samples. In practice, both $l$ and $n$ are large. Alternatively, we can use lazy evaluation to decrease the computation complexity to $O(ln)$ (Leskovec et al., 2007), but it requires an additional hyperparameter to be chosen in advance. Instead of greedily selecting elements in an offline fashion, we adopt the two-pass streaming algorithm of Badanidiyuru et al. (2014), whose complexity is $\tilde{O}(ln)^3$, and generalize it to the knapsack constraint (shown in Alg. 2). In the first pass, we calculate the maximum function value of a single element normalized by its weight, which gives an estimate of the optimal value. In the second pass, we create $O(\frac{1}{\epsilon} \log K)$ buckets and greedily update each of the bucket according to:

$$\Delta_g(e|\mathbb{S}_v) \geq \frac{k(\{e\})(v/2 - g(\mathbb{S}_v))}{K - k(\mathbb{S}_v)}, \tag{8}$$

where each bucket has a different value $v$, and $\Delta_g(e|\mathbb{S}_v) := g(\{e\} \cup \mathbb{S}_v) - g(\mathbb{S}_v)$ is the marginal improvement of submodular function $g$ when adding element $e$ to set $\mathbb{S}_v$. The whole pipeline of the active learning algorithm is shown in Alg. 1. The algorithm gives the following guarantee, which is proven in Appendix.

**Theorem 1.** *Alg. 2 gives a* $\frac{(1-\epsilon)(1-\delta)}{2}$-*approximation guarantee for (6), where* $\delta = \max_{e \in \mathbb{S}} k(\{e\})/K$.

**Proof sketch:**  The criterion (8) we use guarantees that each update we make is reasonably good. The set $\mathbb{S}_v$ stops updating when either the current budget is almost $K$, or any sample in the stream after we reach $\mathbb{S}_v$ does not provide enough marginal improvement. While it is easy to give guarantees when the budget is exhausted, it is unlikely to happen; we use a difference expression between current set $\mathbb{S}_v$ and the optimal set, and prove the gap between the two is under control.

In a practical label acquisition process, the budget we set for each round is usually much larger than the length of the longest sentence in the unlabeled set, making $\delta$ negligible. In our experiments, $\delta$ was around $0.01$.

---

[3]Big $O$ notation ignoring logarithmic factors.

