# OpenReview forum: "Deep Active Learning for Named Entity Recognition"
_ICLR.cc/2018/Conference — Accept (Poster)_

### Official Review · AnonReviewer2 · 2017-11-24
**Active learning for BiLSTM-based NER model**

**Rating:** 7
**Confidence:** 4

**Review:**

Summary:
This paper applies active learning to a deep neural model (CNN-CNN-LSTM) for  named-entity recognition, which allows the model to match state-of-the-art performance with about 25% of the full training data.

Strength:
The paper is relatively easy to follow. Experiments show significant reduction of training samples needed.

Weaknesses:
About half of the content is used to explain the CNN-CNN-LSTM architecture, which seems orthogonal to the active learning angle, except for the efficiency gain from replacing the CRF with an LSTM.

The difference in performance among the sampling strategies (as shown in Figure 4) seems very tiny. Therefore, it is difficult to tell what we can really learn from these empirical results.

Other questions and comments:
In Table 4: Why is the performance of LSTM-LSTM-LSTM not reported for OntoNotes 5.0, or was the model simply too inefficient?

How is the variance of the model performance? At the early stage of active learning, the model uses as few as 1% of the training samples, which might cause large variance in terms of dev/test accuracy.

The SUBMOD method is not properly explained in Section 4. As one of the active learning techniques being compared in experiments, it might be better to formally describe the approach instead of putting it in the appendix.

---

> ### Author Response · Authors · 2017-12-28
> **More experimental results are added to strengthen the empirical evidence**
>
> We would like to thank Reviewer 2 for taking the time to leave a clear and thoughtful review. We have improved the draft per your comments and would like to reply to each specific point in turn:
>
> 1. While the architecture and active learning contributions may appear on a technical level to be orthogonal contributions, in practice they are closely linked. Active learning requires frequent retraining of the model, during which time new annotations cannot be collected; faster training speed makes more frequent training of the model affordable, which in turn allows collection of more informative labels. Whereas the use of recurrent network in word-level encoder and chain CRF in tag decoder have been considered as a standard approach in the literature (from Collobert et al 2011 to more recent papers like Lample et al 2016, Chiu and Nichols 2015, Yang et al 2016), we demonstrate that convolutional word encoder and LSTM decoder provides 4x speedup in training time with very minimal loss of performance in terms of F1 score.
>
> Per your criticism and to reinforce the significance of the speedup, we have implemented standard CNN-LSTM-CRF model and added its training speed to Table 4, so that the magnitude of the speedup could be demonstrated.
>
> 2. We have also added LSTM-LSTM-LSTM experiment on OntoNotes 5.0-English in Table 4. We did not run this experiment in our initial draft since we focus on finding computationally efficient architectures, and it was clear from CoNLL-2003 English experiments that LSTM-LSTM-LSTM is much more computationally expensive than other competitors; but we agree that this result is still informative.
>
> 3. In response to your question about variance, we have replicated the active learning experiments multiple times, and added a learning curve plot with error bars; please refer to Appendix B and Figure 6 of the updated paper. While learning curves from active learning methods are indeed close to each other, there is a noticeable trend that MNLP (our proposal) and BALD outperforms traditional LC in early rounds of data acquisition. Also note that MNLP we propose is much more computationally efficient because it requires only a single forward pass, whereas BALD requires multiple forward passes.
>
> 4. We agree with your point that the SUBMOD method ought to be explained briefly in the main text and not relegated only to the Appendix. Per your criticism, we’ve added a paragraph to section 4 describing the approach.

---

> > ### Comment · AnonReviewer2 · 2018-01-07
> > **Reply**
> >
> > Thank you for such detailed response and for adding the new experiment results. Overall, I believe this kind of empirical results on applying active learning to deep NLP models would be useful to the community.

---

### Official Review · AnonReviewer1 · 2017-11-27
**This paper studies the application of different existing active learning strategies for the deep models for NER. This paper has several strong and weak points listed in the reviews**

**Rating:** 6
**Confidence:** 4

**Review:**

This paper studies the application of different existing active learning strategies for the deep models for NER.

Pros:
* Active learning may be used for improving the performance of deep models for NER in practice
* All the proposed approaches are sound and the experimental results showed that active learning is beneficial for the deep models for NER

Cons:
* The novelty of this paper is marginal. The proposed approaches turn out to be a combination of existing active learning strategies for selecting data to query with the existing deep model for NER.
* No conclusion can be drawn by comparing with the 4 different strategies.

======= After rebuttal  ================

Thank you for the clarification and revision on this paper. It looks better now.

I understand that the purpose of this paper is to give actionable insights to the practice of deep learning. However, since AL itself is a meta learning framework and neural net as the base learner has been shown to be effective for AL, the novelty and contribution of a general discussion of applying AL for deep neural nets is marginal.  What I really expected is a tightly-coupled active learning strategy that is specially designed for the particular deep neural network structure used for NER. Apparently, however, none of the strategies used in this work is designed for this purpose (e.g., the query strategy or model update strategy should at least reflex some properties of deep learning or NER). Thus, it is still below my expectation.

Anyway, since the authors had attempted to improved this paper, and the results may provide some information to practice, I would like to slightly raise my rating to give this attempt a chance.

---

> ### Author Response · Authors · 2017-12-28
> **Our results give actionable insights to the practice of deep learning**
>
> Thank you for taking the time to review our paper. We are glad that you recognize the soundness of the approaches and the experimental evaluation.
>
> We would like to address the reviewer’s main concerns:
>
> 1. Regarding “No conclusion can be drawn by comparing with the 4 different strategies”:
>
> Our experiments demonstrate that active learning can produce results comparable to the state of the art methods while using only 25% of the data. Moreover, we demonstrate that combining representativeness-based submodular optimization methods with uncertainty-based heuristics conferred no additional advantage.
>
> We believe that these are in and of themselves significant conclusions. We agree that the Bayesian approach (BALD) and least confidence approaches (LC and MNLP) produce similar learning curves, but we do not believe that this renders the result inconclusive. To the contrary, parity in performance strongly favors the least confidence approaches owing to their computational advantages; BALD requires multiple forward passes to produce an uncertainty estimate, whereas LC and MNLP require only a single forward pass.
>
> We have added error bars to our learning curve (Appendix B, Figure 6) in our updated paper, and it can be seen that there is a noticeable trend which MNLP (our proposal) and BALD outperform traditional LC in early active learning rounds.
>
> 2. Regarding the novelty of the paper:
>
> In this paper, we explore several methods for active learning. These include uncertainty-based heuristics, which we emphasize in the main text and which yield compelling experimental results. We also include a representativeness-based sampling method using submodular optimization that is reported in the main text and described in detail in the appendix. In the revised version we have included some of this detail in the main body of the paper.
>
> It so happens that the simpler approaches outperform our more technically complex contributions. The tendency not to publish compelling results produced by simple methods creates a strong selection bias that might misrepresent the state of the art and can encourages authors to overcomplicate methodology unnecessarily. While the winning method in our paper doesn’t offer substantial mathematical novelty, we hope that the reviewer will appreciate that our work demonstrates empirical rigor. Our results give actionable insights to the deep learning and NLP communities that are not described in the prior literature.

---

### Official Review · AnonReviewer3 · 2017-11-27
**the ideas are simple but seems to work well empirically**

**Rating:** 6
**Confidence:** 3

**Review:**

This paper introduces a lightweight neural network that achieves state-of-the-art performance on NER. The network allows efficient active incremental training, which significantly reduces the amount of training data needed to match state-of-the-art performance.

The paper is well-written. The ideas are simple, but they seem to work well in the experiments. Interestingly, LSTM decoder seems to have slight advantage over CRF decoder although LSTM does not output the best tag sequence. It'd be good to comment on this.

* After rebuttal
Thank you for your response and revision of the paper. I think the empirical study could be useful to the community.

---

> ### Author Response · Authors · 2017-12-28
> **further analysis of LSTM decoder is provided**
>
> We’d like to thank Reviewer 3 for a thoughtful and constructive review of our paper. We are glad that the reviewer recognized the importance of using a lightweight network to support the incremental training required for our active learning strategy.
>
> The reviewer asks an interesting question; “LSTM decoder seems to have slight advantage over CRF decoder although LSTM does not output the best tag sequence. It'd be good to comment on this.”
>
> Indeed, our greedily decoded tag sequences from the LSTM decoder may not be the best tag sequence with respect to the LSTM’s predictions whereas the chain CRF can output the best sequence with respect to its predictions via dynamic programming. There are a few important points to make here:
>
> 1. We experimented with multiple beam sizes for LSTM decoders, and found that greedy decoding is very competitive to beam search; beam search with size 16 was only marginally better than greedily decoded tags in terms of F1 score. Producing the best sequence is equivalent to setting an arbitrarily large beam width. Out experiments indicate that empirically, increasing the beam width beyond 2 helps only marginally.  This suggests that the intractability of finding best sequence from the LSTM decoder is not a significant practical concern. We added these results, plotting the F1 score vs beam width in Appendix A of the updated paper.
>
> 2. The LSTM decoder is a more expressive model than the chain CRF model since it models long-range dependencies (vs a single step for CRF), and thus the LSTM’s best sequence may be considerably better than the best sequence according to the CRF. So even if we do not get the very best sequence from the LSTM, it can still outperform the CRF. Indeed, we found in our experiments that when the encoder is fixed, our LSTM decoder outperforms our CRF decoder (Table 3 and 4, compare CNN-CNN-CRF against CNN-CNN-LSTM).

---

### Author Response · Authors · 2017-12-28
**General Reply to All Reviewers**

We would like to thank the reviewers for taking the time to review our paper and give thoughtful comments. We are encouraged to see that two reviewers rate the paper as above the acceptance threshold and that all three reviewers recognized the clear demonstration of the empirical benefits of applying active learning when training deep models for named entity recognition. The reviewers also left some insightful critical comments. We have addressed these in the revised draft and responded to each reviewer in the respective threads.

---

### Decision · Program_Chairs · 2018-01-29
**ICLR 2018 Conference Acceptance Decision**

**Decision:**

Accept (Poster)

**Comment:**

The reviewers liked this paper quite a bit. The novelty seems modest and the results are limited to a fairly simple NER task, but there is nothing wrong with the paper, hence recommending acceptance.